# Supplementation of Probiotics in Pregnant Women Targeting Group B Streptococcus Colonization: A Systematic Review and Meta-Analysis

**DOI:** 10.3390/nu14214520

**Published:** 2022-10-27

**Authors:** Daniela Menichini, Giuseppe Chiossi, Francesca Monari, Francesco De Seta, Fabio Facchinetti

**Affiliations:** 1Department of Biomedical, Metabolic and Neural Sciences, International Doctorate School in Clinical and Experimental Medicine, University of Modena and Reggio Emilia, 41124 Modena, Italy; 2Unit of Obstetrics and Gynecology, Mother-Infant Department, University of Modena and Reggio Emilia, 41124 Modena, Italy; 3Institute for Maternal and Child Health “IRCCS Burlo Garofolo”, 34137 Trieste, Italy

**Keywords:** group B streptococcus, probiotics, pregnancy, intrapartum antibiotic prophylaxis, perinatal outcomes, neonatal sepsis

## Abstract

This systematic review and meta-analysis aimed to determine if probiotic supplementation in pregnancy reduced maternal Group B streptococcus (GBS) recto-vaginal colonization in pregnant women at 35–37 weeks of gestation. Electronic databases (i.e., PubMed, MEDLINE, ClinicalTrials.gov, ScienceDirect, and the Cochrane Library) were searched from inception up to February 2022. We included RCTs assessing the effects of probiotic supplementation in pregnancy on GBS recto-vaginal colonization. The primary outcome was GBS-positive recto-vaginal cultures performed at 35–37 weeks of gestation. Secondarily, we evaluated obstetric and short-term neonatal outcomes. A total of 132 publications were identified; 9 full-length articles were reviewed to finally include 5 studies. Probiotic supplementation reduced vaginal GBS colonization: the GBS positive culture rate was estimated at 31.9% (96/301) in the intervention group compared to 38.6% (109/282) in the control group (OR = 0.62, 95% CI 0.40–0.94, I2 4.8%, *p* = 0.38). The treatment started after 30 weeks of gestation and was more effective in reducing GBS colonization (OR 0.41, 95% CI 0.21–0.78, I2 0%, *p* = 0.55). Probiotic administration during pregnancy, namely in the third trimester, was associated with a reduced GBS recto-vaginal colonization at 35–37 weeks and a safe perinatal profile. Whether this new strategy could reduce the exposition of pregnant women to significant doses of antibiotics in labor needs to be evaluated in other trials.

## 1. Introduction

Group B streptococcus (GBS) is an important cause of maternal and neonatal morbidity and mortality worldwide. In Europe, the percentage of women colonized with GBS in pregnancy ranges from 1.5 to 30% and accounts for chorioamnionitis, cystitis, pyelonephritis, bacteremia, fever, and postpartum endometritis [1,2]. The presence of the pathogen in the maternal urinary tract at the time of delivery is the most important risk factor for neonatal GBS infection; alternatively, GBS can reach the amniotic fluid by ascending through the cervix with intact or ruptured membranes [3], especially in the cases of prolonged labor, premature rupture of membranes (PROM), or preterm birth (PTB) [4]. Less frequent GBS-related morbidities are represented by surgical wound infection after cesarean delivery, pelvic abscesses, pelvic septic thrombophlebitis, and osteomyelitis [5]. Approximately 98% of colonized newborns are asymptomatic while the early-onset symptomatic forms have a 1–3% incidence with a 50–60% neonatal mortality [6,7].

GBS is detected with universal culture screening at 35–37 weeks’ gestation; currently, antibiotic prophylaxis in active labor is the most effective intervention to counteract early neonatal infections in GBS-positive women [8]. Antibiotic prophylaxis is also effective when administered to women with risk factors for GBS colonization (i.e., labor <37 weeks, amniotic membrane rupture for ≥18 h, or intrapartum T > 38°C) and unknown GBS status [9].

However, the widespread use of intrapartum antibiotics likely affects the biodiversity of maternal and neonatal microbiota and is associated with mother and infant gut microbiota dysbiosis [10,11]. The vaginal microbiota has indeed been recognized as a novel factor by which maternal stress and perturbations may contribute to reprogramming the developing brain of the offspring, predisposing individuals to neurodevelopmental disorders [12].

Moreover, intrapartum antibiotic prophylaxis (IAP) may secondarily decrease the susceptibility to penicillin or ampicillin, the agents of choice used to prevent GBS disease [13,14]. Therefore, several strategies have been tested as alternatives. A small study showed that intrapartum vaginal flushing with chlorhexidine was as effective as ampicillin in preventing GBS transmission to neonates and also reduced the rate of neonatal *E. coli* colonization [15]. The WHO launched the first GBS maternal immunization program to develop a GBS vaccine but the results are not available yet [16]. Natural antibacterial phytochemicals (i.e., Carvacrol) have been shown to compromise the cell membrane integrity by inducing changes that lead to leakage of cytoplasmic contents such as lactate dehydrogenase enzymes and nucleic acids, demonstrating an additive–synergistic effect with clindamycin or penicillin [17]. In addition, plant-based compounds were used to inhibit the virulence properties and gene expression [18] of Streptococcus species; indeed, promising results have been reported regarding the synergistic effects of citral (citrus oil with anti-inflammatory and bactericidal properties) and phloretin (a polyphenolic chalcone that has many interesting biological properties, including inhibition of Gram-positive and Gram-negative bacteria) to combat the virulence of *Streptococcus* [19]. Finally, probiotics were studied to reduce GBS colonization rates at 35–37 weeks of gestation to prevent neonatal infections.

Probiotics are live microorganisms that, when administered in adequate amounts, confer a health benefit to the host [20]. Their supplementation is increasingly widespread and accepted globally due to their documented health benefits [21,22].

Research studies have proven that women with higher vaginal colonization of lactobacilli are more likely to have no detectable vaginal GBS [23,24,25]. Indeed, probiotics have the potential to maintain vaginal homeostasis through the occupation of niches that impede the expansion of other bacteria and the establishment of biofilms, the increase in lactic acid and production of other antimicrobial compounds, and the regulation of the local cervicovaginal mucosal immune responses [26,27,28]. Moreover, no major safety concerns were reported for probiotics [29] and a recently published systematic review and meta-analysis stated that probiotics and prebiotics in pregnancy and lactation were safe. Only one study that administered *Lactobacillus rhamnosus* and *L. reuteri* showed a higher risk of vaginal discharge and changes in stool consistency, but overall, no serious health concerns to the mother or infant have been raised regarding probiotic and prebiotic use [30].

On these grounds, we conducted a systematic review and meta-analysis to summarize the available evidence on the effects of probiotic supplementation to decrease maternal GBS recto-vaginal colonization.

## 2. Materials and Methods

### 2.1. Search Strategy

The review protocol was established by two investigators (G.C. and D.M.) before the commencement of the study and was registered with the PROSPERO International Prospective Register of Systematic Reviews (registration no. 184589). The electronic databases MEDLINE, ClinicalTrials.gov, PROSPERO, and the Cochrane Central Register of Controlled Trials were searched from the inception of each database until February 2022 using the following terms: ‘GBS’, ‘group B streptococcus’, ‘colonization’, ‘probiotics’, ‘recto-vaginal colonization’, ‘GBS colonization’, and ‘randomized trial’. All manuscripts were reviewed for pertinent references. No language restrictions were applied.

### 2.2. Study Selection

Selection criteria included RCTs that evaluated the effects of probiotic supplementation in pregnancy on GBS recto-vaginal colonization. We included RCTs involving pregnant women receiving probiotics. The primary outcome was GBS-positive recto-vaginal cultures performed at 35–37 weeks’ gestation. Secondarily, we evaluated obstetric outcomes: preterm birth (PTB), preterm rupture of membranes (PROM), chorioamnionitis, and neonatal outcomes (neonatal infection and neonatal intensive care unit (NICU) admission).

### 2.3. Data Extraction and Risk-of-Bias Assessment

Data from each eligible study were extracted without modification of original data onto custom-made data collection forms. A two-by-two table was used to calculate the relative risk (OR). The summary measures were reported as OR with 95% CI; between-study heterogeneity was accounted for using random-effects meta-analyses. Subgroup analyses were performed according to the positive or unknown GBS baseline according to the gestational age at beginning of the treatment with probiotics (after 30 weeks or before 30 weeks) and to the duration of the treatment (less or more than 12 weeks). Data analysis was performed using Stata 15.1 (StataCorp, College Station, TX, USA).

We assessed the risk of bias in each included study using the criteria outlined in the *Cochrane Handbook for Systematic Reviews of Interventions* [31]. Seven characteristics related to the risk of bias were assessed in each included trial because there is evidence that these issues are associated with biased estimates of treatment effect: (1) random sequence generation; (2) allocation concealment; (3) blinding of participants and personnel; (4) blinding of outcome assessment; (5) incomplete outcome data; (6) selective reporting; and (7) other bias. Review authors’ judgments were categorized as ‘low risk’, ‘high risk’, or ‘unclear risk’ of bias [31]. Publication bias was evaluated using a funnel plot. Our study was exempt from IRB approval because it collected and integrated publicly available research. The review was reported according to the Preferred Reporting Items for Systematic Reviews and Meta-Analyses (PRISMA) statement (see the Appendix A).

## 3. Results

The flow diagram of the electronic search details and selection process is shown in Figure 1. A total of 132 publications were identified; of these, 123 were excluded according to the title or the abstract, while 9 full-length articles matched the inclusion criteria. After revision, three studies were excluded because they were non-RCTs and one was excluded because only the protocol was available. Thus, five eligible RCTs were finally included in the analysis. The main features of the included studies are summarized in Table 1.

As reported in Figure 2, most of the included studies were considered to have a low or unclear risk of bias. The blinding of participants and personnel was the most frequent bias among the included studies. All of the studies were exempt from selection bias thanks to the randomization, as well as from reporting bias.

The population was Caucasian in three studies [33,35,36] while one study also included Hispanic and other ethnicities [34]; the remnant one was conducted on Asian women [32]. The sample size ranged from 34 to 151 pregnant women. All of the included studies tested oral supplementation with probiotics. Probiotic strains were the same in four of the studies (*Lactobacillus rhamnosus GR-1* and *Lactobacillus reuteri RC-14 L*) [32,33,34,35], while *Lactobacillus jensenii Lbv116*, *Lactobacillus crispatus Lbv88*, *Lactobacillus rhamnosus Lbv96*, and *Lactobacillus gasseri Lbv150* were used in the remnant study [36]. Doses ranged from 1 ×108 CFU to 5.4 ×109 CFU daily. The duration of supplementation ranged from 2 to 12 weeks (Table 1). While three studies investigated women known to have GBS-positive cultures [32,33,36], the remnants included women with an unknown GBS status [34,35].

Probiotic supplementation caused a drop in vaginal GBS colonization: the GBS-positive culture rate was 31.9% (96/301) in the intervention group compared to 38.6% (109/282) in the placebo group (OR = 0.62, 95% CI 0.40–0.94, I2 4.8%, *p* = 0.38; Figure 3).

This positive result also was confirmed among women with a GBS-positive baseline that encountered a significant conversion to negative culture after probiotic treatment (OR = 0.41, 95%CI 0.21–0.78, I2 0%, *p* = 0.55; Figure 4).

The subgroup analysis showed that if the treatment was started after 30 weeks of gestation, it was more effective in reducing GBS colonization (OR 0.41, 95% CI 0.21–0.78, I2 0%, *p* = 0.55; Figure 5).

The duration of the treatment (less or more than 12 weeks) did not seem to alter the effect on GBS colonization because the stratified group showed positive trends toward the protective effect of probiotics (Figure 6).

The secondary outcomes were not meta-analyzed because they were reported in only a few studies; these outcomes are summarized in Table 2. Ming-Ho et al. [32] and Sharpe et al. [35] described fewer NICU admissions as well as lower rates of clinical chorioamnionitis and neonatal infections when probiotics were prescribed to pregnant mothers, although significance was not reached. No differences in intrapartum fever, preterm birth, or neonatal infections were reported in the other studies. Of note, none of the studies reported cases of adverse effects related to probiotics, neither for the mothers nor the babies.

No risk of publication bias was detected according to funnel plot (Figure 7).

## 4. Discussion

GBS colonizes approximately 20% of pregnant women and represents the most important risk factor for neonatal early-onset sepsis (EOS) with a high rate of morbidity and mortality.

This systematic review and meta-analysis was conducted to determine if probiotic supplementation in pregnancy reduced maternal GBS recto-vaginal colonization in pregnant women at 35–37 weeks of gestation.

We found that women receiving probiotic supplementation in pregnancy, when compared to those receiving placebo, had lower GBS positive recto-vaginal cultures at universal screening performed at 35–37 weeks gestation. These findings also were confirmed among known GBS-positive women who reached a higher conversion to a negative culture [32,36]. A subanalysis showed that this effect was amplified when treatment began after the 30th week of gestation, meaning that proximity to delivery could play a key role, while the long duration of the treatment did not improve the effects.

Such an effect was observed across all studies and was independent of the study effect size as indicated by the low between-study heterogeneity in the treatment effect (I-squared of 4.8%) and the low between-study variance (tau-squared of 0.01).

The composition of probiotics in primary studies included *Lactobacillus* spp., which have shown anti-GBS activity “in vitro” [37]. Indeed, *Lactobacillus* cells have been demonstrated to be able to interact and aggregate with *Streptococcus* cells and kill the GBS, underlining the importance of bacterial co-aggregation as an antimicrobial mechanism against pathogens [37]. Afterward, *Lactobacilli* compete with Streptococcus for adhesion to vaginal mucosa cells and nutrients and produce antimicrobial substances (hydrogen peroxide, lactic acid, and bacteriocins) that affect GBS replication; they can also counteract other pathogens such as *C. vaginalis* and *N. gonorrheae* as demonstrated in bacterial co-aggregation studies [38,39,40].

In three of the included studies, probiotic administration did not significantly reduce GBS recto-vaginal colonization [33,34,35]. The short duration of the intervention (3 weeks) may account for such results in the study by Olsen et al. [33], while the probiotic composition, compliance, and population baseline characteristics may have played a role in the studies by Sharpe et al. [35] and Aziz et al. [34].

Regarding safety concerns, probiotics are generally considered safe and well tolerated. Current data suggest that probiotic supplementation is rarely systemically absorbed when used by healthy individuals. One meta-analysis of several randomized controlled trials conducted with women during the third trimester did not report an increase in adverse neonatal outcomes [41]. We confirmed these findings supporting maternal probiotic administration, which did not worsen short-term neonatal health (NICU admission or sepsis). Furthermore, according to a recently published meta-analysis and systematic review, probiotic products have other clinical benefits during or after pregnancy such as preventing or treating gestational diabetes [42], mastitis [43], preterm birth [44], and infantile atopic dermatitis [45].

Therefore, these products may contribute to improving the health of pre-pregnant, pregnant, and postpartum patients and their children in specific situations, and their benefits may outweigh the documented minimal risks.

Our meta-analysis demonstrated that probiotic supplementation was also associated with a significant reduction in emergency cesarean sections [33]. However, the trial was not equipped for this secondary outcome and the finding has not been confirmed in another RCT [36].

Interestingly, another study reported that prenatal probiotics significantly reduced the incidence of bacterial vaginosis, increased colonization with vaginal *Lactobacillus* and intestinal *Lactobacillus rhamnosus*, altered immune markers in serum and breast milk, and improved maternal glucose metabolism, resulting in significantly higher counts of *Bifidobacterium* and *Lactococcus lactis* (healthy intestinal flora) in neonatal stool [46].

It is nowadays recognized that the maternal microbiota influences the colonization in the infant. Recent studies suggested that this mechanism begins before delivery during intrauterine life. Indeed, during gestation, the fetus can encounter microorganisms of maternal origin. In fact, fragments of bacterial DNA have been found in the umbilical cord, in the amniotic fluid, and even in the meconium [47].

Their presence is made possible by the fact that during gestation, the maternal gut becomes more permeable, which favors bacterial translocation. Commensal microbes translocate from the maternal gut to the placenta or fetal gut during pregnancy. These microbes impact the development of fetal immunity via various mechanisms including epigenetic changes, the release of short-chain fatty acids, and alteration of the cytokine environment. This aspect is of fundamental importance because if the mother is in conditions of eubiosis, the contact of the fetus with the correct bacterial strains will create a very favorable condition for the newborn gut. If, on the other hand, the mother has an altered microbiota, it is possible to witness the passage from the maternal bloodstream through the placenta to the fetus of different bacterial strains that could lead to greater exposure of the newborn to diseases [48].

This reinforces the importance of maintaining a healthy maternal microbiota not only in the proximity of childbirth, but also throughout the entire pregnancy and even during breastfeeding thanks to healthy lifestyles and the use of probiotics.

A recent systematic review and meta-analysis [49] that also included non-randomized and quasi-experimental clinical studies reported the efficacy of probiotic intervention in reducing the rate of GBS-positive women, although with less strength of the evidence (few placebo-controlled studies). Moreover, our evaluation, which included RCTs only, provided additional information. Indeed, the efficacy of probiotics seems to be related to third-trimester supplementation with respect to treatment implemented long before parturition. Finally, we also provided some insights into neonatal health.

Supporting these promising data on probiotics and neonatal health, a recent multi-center study was conducted on infants (up to 32 weeks’ gestation) admitted to 289 neonatal intensive care units (NICUs) receiving probiotics during the first postnatal days. Several adverse outcomes were evaluated: necrotizing enterocolitis (NEC), bloodstream infections, meningitis, and death. The authors reported a decrease in the odds of NEC and death but concluded that little is known about the doses of particular strains and the mechanism of action that determine which treatment produces the maximum safety and efficacy [50].

The strengths of our study included the comprehensive search strategy of including only RCTs with no language restrictions and the low degree of heterogeneity in the included studies. The limitations primarily related to the low number of studies included in the meta-analysis and the quality of the studies, as some of them were judged to have an unclear risk of bias. In particular, the study by Aziz et al. [34], which had the highest risk of bias, was also the most numerous one, thus it primarily drove the others included in the meta-analysis. Moreover, since probiotics are not treated as medicines, the different strains at different dosages and different routes of administration are not well regulated by governing agencies in most countries. In addition, compliance with treatment was not adequately addressed across the surveys, as only Olsen et al. [33] identified that fully compliant women had a significant increase in the quantity of vaginal commensal bacteria.

## 5. Conclusions

Preventing maternal GBS colonization has an important impact on the health of the newborn because it may avoid the 1–3% of early-onset symptomatic forms [6,7]. It also has an impact on women’s exposure to IAP, which has potential perinatal microbiological sequelae of exposure for the mother and the newborn. Thus, primary prevention strategies for GBS colonization are increasingly urgent; probiotics, with their antagonistic activities against GBS, are promising. This systematic review and meta-analysis demonstrated that the administration of probiotics during pregnancy, namely in the third trimester, was associated with a reduced GBS recto-vaginal colonization at 35–37 weeks and a safe perinatal profile. Probiotics also may be useful to counteract GBS colonization when it is already established, showing a considerable negativization rate in GBS-positive women. Whether this new strategy could reduce the exposure of pregnant women to significant doses of antibiotics in labor needs to be further investigated. Future double-blind randomized controlled trials with larger and more diverse samples are required.

## Figures and Tables

**Figure 1 nutrients-14-04520-f001:**
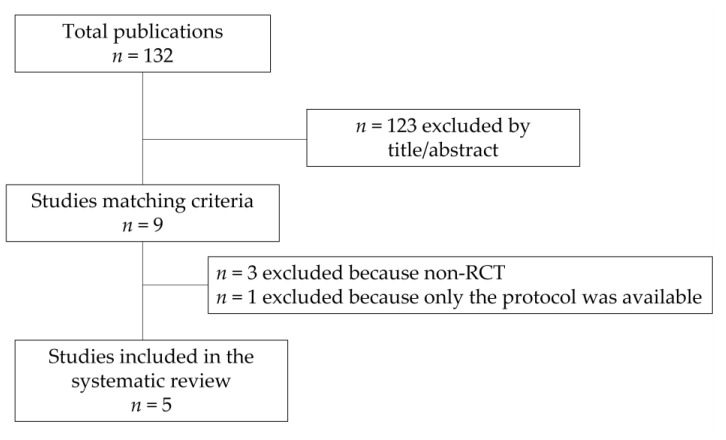
Flow diagram of the study search process.

**Figure 2 nutrients-14-04520-f002:**
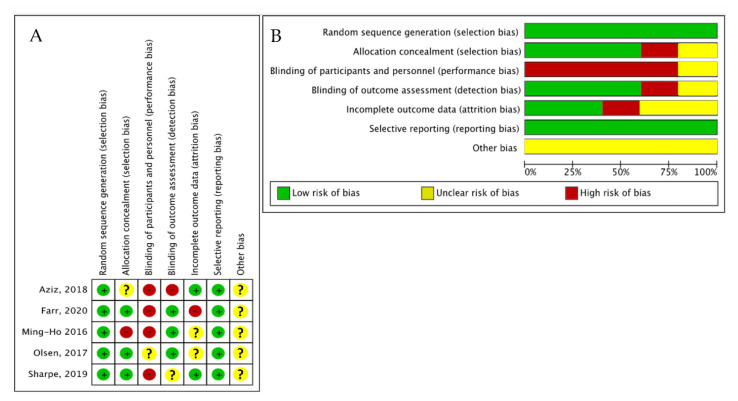
Assessment of bias risk. (**A**) Summary of risk of bias for each trial. A plus sign indicates a low risk of bias; a minus sign indicates a high risk of bias; a question mark indicates an unclear risk of bias. Aziz, 2018 [34], Farr, 2020 [36], Ming-Ho, 2016 [32], Olsen, 2017 [33], Sharpe, 2019 [35]. (**B**) Risk of bias graph for each risk of bias item presented as percentages across all included studies.

**Figure 3 nutrients-14-04520-f003:**
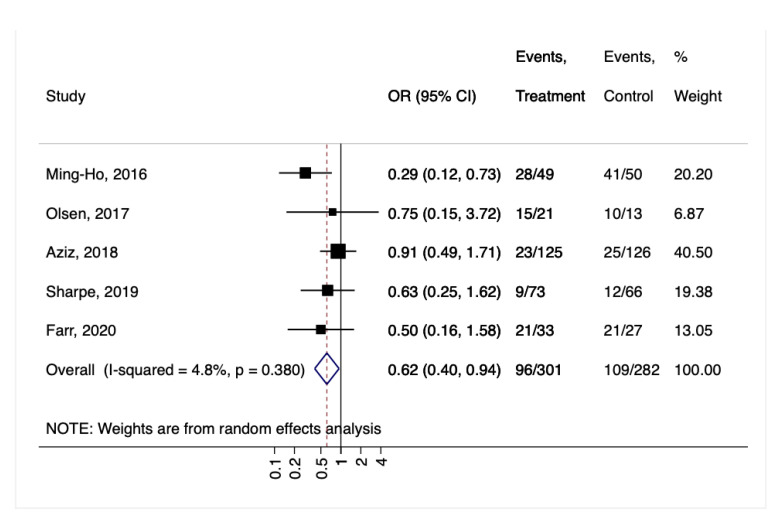
Forest plot for the GBS colonization. Ming-Ho, 2016 [32], Olsen, 2017 [33], Aziz, 2018 [34], Sharpe, 2019 [35], Farr, 2020 [36].

**Figure 4 nutrients-14-04520-f004:**
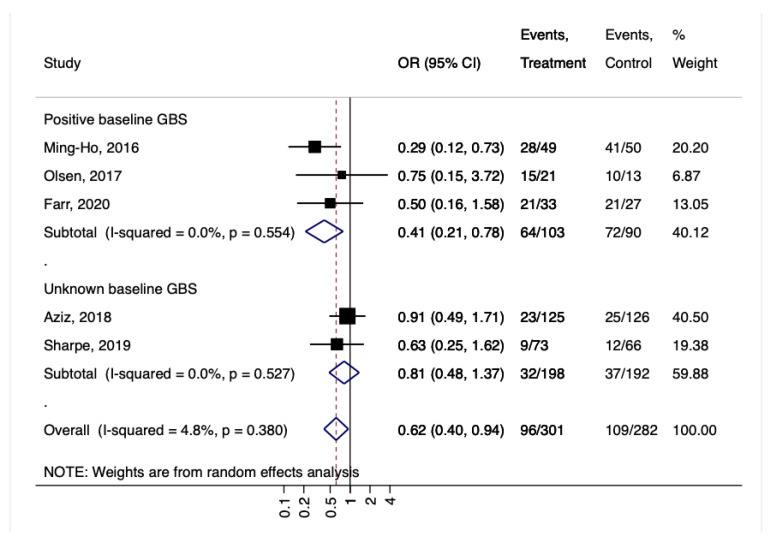
Forest plot for the GBS colonization according to positive or unknown GBS baseline. Ming-Ho, 2016 [32], Olsen, 2017 [33], Farr, 2020 [36], Aziz, 2018 [34], Sharpe, 2019 [35].

**Figure 5 nutrients-14-04520-f005:**
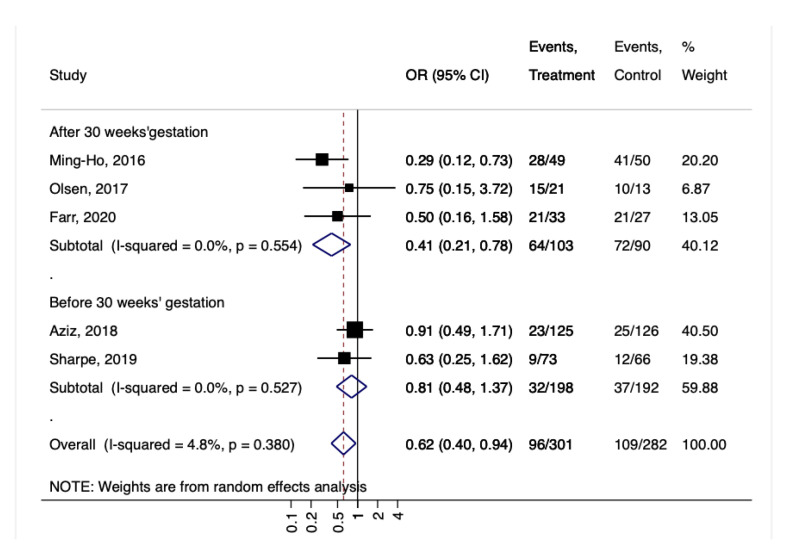
Forest plot for the GBS colonization according to the gestational age at beginning of the treatment with probiotics (after 30 weeks or before 30 weeks). Ming-Ho, 2016 [32], Olsen, 2017 [33], Farr, 2020 [36], Aziz, 2018 [34], Sharpe, 2019 [35].

**Figure 6 nutrients-14-04520-f006:**
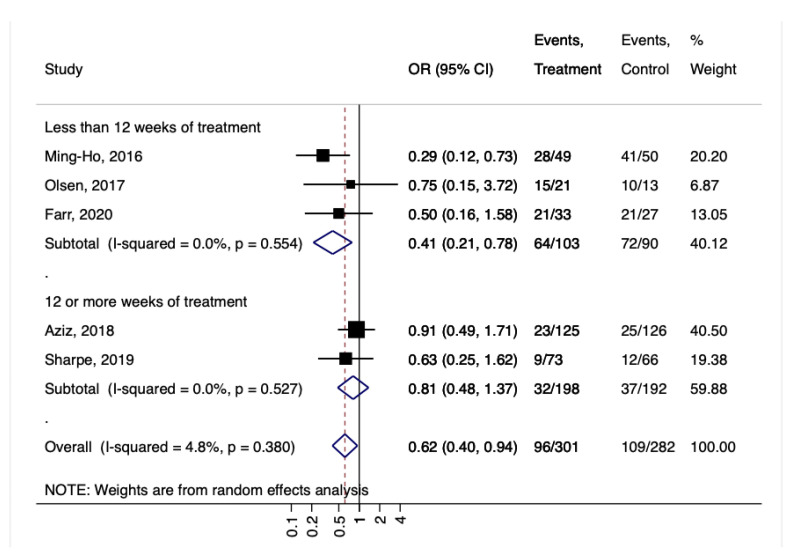
Forest plot for the GBS colonization according to the duration of treatment with probiotics (less than 12 weeks or more than 12 weeks). Ming-Ho, 2016 [32], Olsen, 2017 [33], Farr, 2020 [36], Aziz, 2018 [34], Sharpe, 2019 [35].

**Figure 7 nutrients-14-04520-f007:**
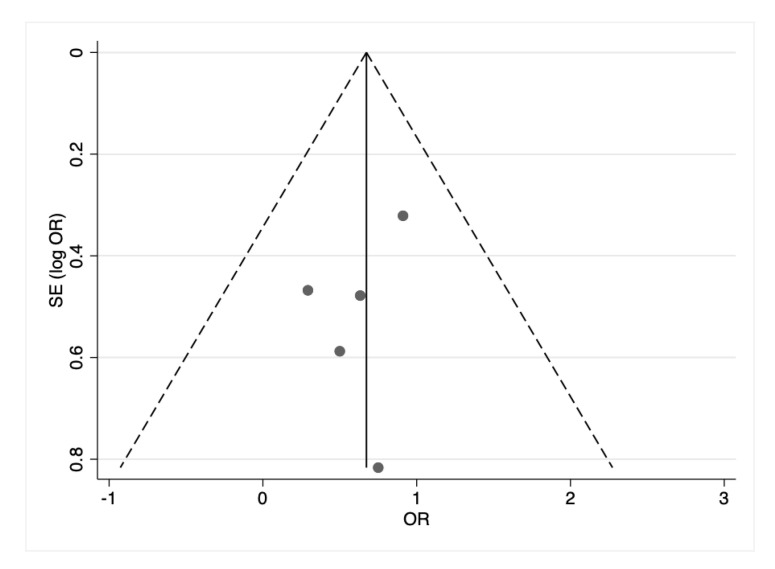
Publication bias.

**Table 1 nutrients-14-04520-t001:** Study and population characteristics.

Study	Country	Ethnicity	Probiotic/ Placebo	Intervention	Doses	Time
Ming-Ho, 2016 [32]	China	Asian	49/50	*L. rhamnosus GR-1* and *L. reuteri RC-14*	2 × 10^9^ CFU/day from 35–37 weeks until delivery	2 weeks
Olsen, 2017 [33]	Australia	Caucasian	21/13	*L. rhamnosus GR-1* and *L. reuteri RC-14*	1 × 10^8^ CFU (108 viable strains) for three weeks/until delivery	3 weeks
Aziz, 2018 [34]	USA	Caucasian Hispanic Other	125/126	*L. rhamnosus GR-1* and *L. reuteri RC-14*	5.4 × 10^9^ CFU daily in capsule started at 28 weeks	12 weeks
Sharpe, 2019 [35]	Canada	Caucasian	73/66	*L. rhamnosus GR-1* and *L. reuteri RC-14*	5 × 10^9^ daily started at 23–25th week	12 weeks
Farr, 2020 [36]	Austria	Caucasian	33/27	*L. jensenii Lbv116; L. crispatus Lbv88;* *L. rhamnosus Lbv96; L. gasseri Lbv150*	4 × 10^9^ CFU daily oral intake started between 32–36 weeks	2 weeks

CFU: colony-forming unit.

**Table 2 nutrients-14-04520-t002:** Secondary outcomes in the probiotic and control/placebo groups.

Study	N	Maternal	Labor and Delivery	Intervention
Ming-Ho, 2016 [32]	99	Intrapartum fever: Placebo: 0/50 Probiotic: 1/49 (2.0%)	N/A	NICU admission Placebo: 0/50 Probiotic: 1/49 (2.0%)
Olsen, 2017 [33]	34	PTB Control: 0/13 Probiotic: 0/21	Emergency CS Control: 5/13 (38.5%) Probiotic: 0/21 *	Neonatal allergies ª Control:0/13 Probiotic: 0/21
Aziz, 2018 [34]	251	PTB Placebo: 3/121 (2.5%) Probiotic: 4/116 (3.5%)	Chorioamnionitis Placebo: 4/116 (3.5%) Probiotic: 5/113 (4.4%)	Neonatal infections Placebo: 2/121 (1.7%) Probiotic: 4/115 (3.5%)
Sharpe, 2019 [35]	139	N/A	Intrapartum infections Placebo: 3/56 (5.3%) Probiotic: 4/57 (7.0%)	NICU admission Placebo: 3/56 (5.3%) Probiotic: 0/57
Farr, 2020 [36]	60	PTB Placebo: 1/41 (2.4%) Probiotic: 4/41 (9.8%)	Cesarean section Placebo: 22/41 (53.7%) Probiotic: 22/41 (53.7%)	Neonatal sepsis Placebo: 0/41 Probiotic: 0/41

ª Asthma, rhinitis, or eczema. * Significant values (*p* < 0.05).

## Data Availability

The data are available upon reasonable request from the corresponding author (D.M.; email: daniela.menichini91@gmail.com).

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
