# Peer review of "Supplementation of Probiotics in Pregnant Women Targeting Group B Streptococcus Colonization: A Systematic Review and Meta-Analysis"

_nutrients, 2022, doi:10.3390/nu14214520_

Round 1

Reviewer 1 Report

This is a systematic review and a meta-analysis of 5 RCTs on probiotic supplementation for prevention of GBS colonisation in pregnancy. The methodology is sound and the paper is well-written. I only have a couple of comments:

1) The study groups in the trials could be described in more detail. It is not clear from table 1 whether all RCTs studied oral probiotic supplementation. This is specified only for one study. In general, probiotic supplementation is far from being a uniform intervention. Different strains at different dosages and different routes of administration have been studied. Moreover, many of these preparations are not well regulated (e.g. in terms of actual probiotic content) since they are not treated as medicines by governing bodies in most countries. This makes drawing firm conclusions on efficacy of probiotic supplementation on GBS colonisation difficult and should be mentioned among the limitations.

2) Four of the five RCTs included in the meta-analysis were negative (with CIs crossing one). The overall effects were, therefore, primarily driven by one study, which is the one considered at highest risk of bias. I believe this should be mentioned at least in the discussion.

Author Response

This is a systematic review and a meta-analysis of 5 RCTs on probiotic supplementation for prevention of GBS colonisation in pregnancy. The methodology is sound and the paper is well-written. I only have a couple of comments:

  • The study groups in the trials could be described in more detail. It is not clear from table 1 whether all RCTs studied oral probiotic supplementation. This is specified only for one study. In general, probiotic supplementation is far from being a uniform intervention. Different strains at different dosages and different routes of administration have been studied. Moreover, many of these preparations are not well regulated (e.g. in terms of actual probiotic content) since they are not treated as medicines by governing bodies in most countries. This makes drawing firm conclusions on efficacy of probiotic supplementation on GBS colonization difficult and should be mentioned among the limitations.

We agree with the reviewer. A more detailed description on the study groups have been added to enrich what is reported in table 1 (please see lines 133-136). We also add a paragraph in the discussion section where we acknowledged among the limitation of this study that the different strains at different dosages and different routes of administration are not well regulated by governing agencies in most countries (please see lines 221-223).

  • Four of the five RCTs included in the meta-analysis were negative (with CIs crossing one). The overall effects were, therefore, primarily driven by one study, which is the one considered at highest risk of bias. I believe this should be mentioned at least in the discussion.

We thank the reviewer for the comment. As suggested, we acknowledged among the limitations that the study by Aziz et al., which is the largest, thus primarily driving the others, is the one at highest risk of bias. (Please see lines 219-220).

However, we would like to point out that the 4 trials, beside the Aziz et al. one, have all the same trend toward a “protective” effect of Probiotics against GBS colonization.

Indeed, Aziz's study, which is the most numerous and thus it drives the others included in the meta-analysis, is the one with a milder effect of probiotics.

Reviewer 2 Report

In the introduction section: The author should include other studies as references where plant-based compounds were used to inhibit the virulence properties and gene expression of Streptococcus species. Also, include the importance of probiotics over natural-based compounds in Streptococcus colonization. 

Suggestions to add references: Adil, M., Baig, M. H., & Rupasinghe, H. V. (2019). Impact of citral and phloretin, alone and in combination, on major virulence traits of Streptococcus pyogenes. Molecules24(23), 4237.

Khan, Rosina, et al. "In vitro and in vivo inhibition of Streptococcus mutans biofilm by Trachyspermum ammi seeds: an approach of alternative medicine." Phytomedicine 19.8-9 (2012): 747-755.

Author Response

In the introduction section: The author should include other studies as references where plant-based compounds were used to inhibit the virulence properties and gene expression of Streptococcus species. Also, include the importance of probiotics over natural-based compounds in Streptococcus colonization. 

Suggestions to add references: Adil, M., Baig, M. H., & Rupasinghe, H. V. (2019). Impact of citral and phloretin, alone and in combination, on major virulence traits of Streptococcus pyogenes. Molecules24(23), 4237.

Khan, Rosina, et al. "In vitro and in vivo inhibition of Streptococcus mutans biofilm by Trachyspermum ammi seeds: an approach of alternative medicine." Phytomedicine 19.8-9 (2012): 747-755.

We thank the reviewer for the comment. As suggested, we enriched the introduction by adding studies on plant-based compounds to counteract the Streptococcus species. Please see lines 61-66.